# Vitamin D Status and Risk of All-Cause and Cause-Specific Mortality in Osteoarthritis Patients: Results from NHANES III and NHANES 2001–2018

**DOI:** 10.3390/nu14214629

**Published:** 2022-11-03

**Authors:** Jing Wang, Jiayao Fan, Ye Yang, Sara Moazzen, Dingwan Chen, Lingling Sun, Fan He, Yingjun Li

**Affiliations:** 1Department of Epidemiology and Health Statistics, School of Public Health, Hangzhou Medical College, Hangzhou 310053, China; 2School of Public Health and The Second Affiliated Hospital, Zhejiang University School of Medicine, Hangzhou 310030, China; 3Molecular Epidemiology Research Group, Max-Delbrück-Centrum für Molekulare Medizin in der Helmholtz-Gemeinschaft, 13125 Berlin, Germany; 4Primary Health Research Center of Zhejiang Province, Hangzhou Medical College, Hangzhou 310053, China; 5Department of Orthopaedics, The Second Affiliated Hospital, Zhejiang University School of Medicine, Hangzhou 310009, China; 6Zhejiang Provincial Center for Disease Control and Prevention, Hangzhou 310051, China

**Keywords:** Vitamin D, osteoarthritis, mortality, NHANES

## Abstract

Objectives: The role of Vitamin D (VD) in calcium balance and bone health makes VD a vital factor in osteoarthritis (OA). Studies that have evaluated the effect of VD on OA patients have mainly been performed on a short-term basis. In this analysis, we aimed to evaluate whether VD was associated with mortality, a long-term outcome, in OA patients. Methods: Participants with self-reported OA from NHANES III and NHANES 2001–2018 were included. Associations of 25(OH)D concentrations with mortality risk were assessed continuously using restricted cubic splines and by categories (i.e., <25.0, 25.0–49.9, 50.0–74.9, and ≥75.0 nmol/L) using the Cox regression model. Sensitivity and stratified analyses were performed to evaluate the robustness of the results. Results: A total of 4570 patients were included, of which 1388 died by 31 December 2019. An L-shaped association was observed between 25(OH)D concentrations and all-cause mortality, whereas an inverse association was found for cardiovascular disease (CVD) mortality. The adjusted hazard ratios (95% confidence intervals) across four categories were 1.00 (reference), 0.49 (0.31, 0.75), 0.45 (0.29, 0.68), and 0.43 (0.27, 0.69) for all-cause mortality and 1.00 (reference), 0.28 (0.14, 0.59), 0.25 (0.12, 0.51), and 0.24 (0.11, 0.49) for CVD-specific mortality; no significant associations were found for cancer-specific mortality. Similar results were observed when stratified and sensitivity analyses were performed. Conclusions: Compared with patients with insufficient or deficient serum 25(OH)D, those with sufficient 25(OH)D concentrations had a lower risk of all-cause and CVD mortality, supporting a beneficial role of VD on a long-term basis.

## 1. Introduction

Osteoarthritis (OA) is a common musculoskeletal disorder that affected approximately 527.8 million people worldwide in 2019 [1]. In the US, more than 32.5 million adults have OA, and it has become a dominant cause of pain and disability [2]. In addition, OA can impact both the physical and mental health of individuals, and such a prevalent disease has manifested in tremendous societal and personal expenses [2,3]. The pathogenesis and progression of OA are multifactorial, including the joint structures, non-modifiable factors (age, gender, etc.), and modifiable factors such as obesity and Vitamin D status [4].

As a steroidal hormone, Vitamin D produces a variety of biological effects on many target tissues [5]. The well-established roles of Vitamin D in the calcium metabolism of bone make Vitamin D a highly influential factor that can be used for the prevention and management of OA [5]. Although a conclusive result has not been reached, several randomized control trials indicated that Vitamin D supplementation was possible to alleviate pain and improve joint function in knee OA patients, especially in those with a low serum Vitamin D level (<50 nmol/L) [6,7,8]. Such benefits might also be relevant to increasing outdoor activity and mental health in OA patients, thus contributing to better survival in terms of quality and quantity [9]. Additionally, Vitamin D has been shown to have benefits in many non-skeletal diseases, such as cardiovascular disease (CVD), diabetes, and cancers, via its effects of reducing inflammation, anti-proliferative activity, anti-oxidative activity, etc. [10]. In a general population, a study including 365,530 subjects from UK Biobank demonstrated that a higher level of serum 25(OH)D (a stable indicator of the Vitamin D status) was nonlinearly related with a decrease in the risk of all-cause, CVD-specific, and cancer-specific mortality [11]. A similar association was reported for all-cause and CVD-specific mortality but not for cancer-specific mortality when the population was restricted to diabetic patients [12].

Although the benefit of Vitamin D in mortality reduction has been reported in the general population and diabetic patients, there have only been a few studies that have focused on such a relationship for OA patients so far. Therefore, in this study, we aimed to evaluate whether serum 25(OH)D concentrations were associated with all-cause and cause-specific mortality in OA patients from NHANES III and the newly released NHANES 2001–2018 database.

## 2. Materials and Methods

### 2.1. Study Population

National Health and Nutrition Examination Survey (NHANES), sponsored by National Center for Health Statistics (NCHS), is a large-scale and periodic program that harvests nationally representative health-related data of non-institutionalized citizens in the United States. Details of survey design and data files are publicly available on https://www.cdc.gov/nchs/nhanes/ (accessed on 31 August 2022). The ethics protocol was approved by the Research Ethics Review Board of NCHS, and informed consent was signed by all recruited participants.

Here, the data from NHANES III (from 1988 to 1994) and NHANES 2001–2018 were used in our analysis, and only respondents (aged > 18 years) with self-reported OA were included in our analysis. OA patients were identified by answering “Yes” to the question “Has a doctor ever told you that you had arthritis?” and then selecting “Osteoarthritis” to the question “Which type of arthritis was it?”. As a common chronic disease, self-reported information was considered to be reliable in most instances; an 85% concordance between self-reported OA and clinically well-defined OA was exhibited in a previous study [13]. At first, 5735 OA patients were identified. After excluding those who did not have 25(OH)D concentrations measurements (647), mortality (31), or other relevant covariates (487), 4570 individuals were eventually included in our study (Figure 1).

### 2.2. Measurement of Serum 25(OH)D

Note that there exist methodological discrepancies in measuring serum 25(OH)D between survey cycles in NHANES. More specifically, the DiaSorin RIA kit (Stillwater, MN, USA) was applied in NHANES III and NHANES 2001–2006, while a standardized liquid chromatography–tandem mass spectrometry (LC-MS/MS) method became an alternative starting from the 2007–08 cycle. We performed standardization using the regression method, referring to the Analysis Guidance Manual provided by NHANES, to unify all serum 25(OH)D data to be LC-MS/MS equivalent for subsequent analyses. The specific conversion method can be accessed at https://wwwn.cdc.gov/nchs/nhanes/vitamind/analyticalnote.aspx (accessed on 31 August 2022). Moreover, for analytes with serum 25(OH)D concentrations below the lower limit of detection (LLOD), the imputed fill value, *LLOD*/*sqrt*(2), was placed in the analytic result field.

### 2.3. Mortality Ascertainment

The corresponding mortality information for each participant was acquired by a linkage to the National Death Index (NDI) up to 31 December 2019. The ICD-10 was applied to determine the underlying causes of death [14]. The primary mortality outcomes considered in our study included all-cause mortality, CVD-specific mortality (codes I00–I09, I11, I13, and I20–I51), cancer-specific mortality (codes C00–C97), as well as other-cause mortality. More information regarding linkage methodology and analytic considerations was elucidated elsewhere [15].

### 2.4. Covariates

A number of variables were identified as potential confounders in our analysis in the following main aspects: (1) Demographic information: age (years); gender (male, female); race/ethnicity (non-Hispanic white, non-Hispanic black, Mexican American, others); education level (below, equivalent, and above high school); family poverty income ratio (PIR; ≤1, 1< to ≤3, >3). (2) Health status: the body mass index (BMI) was classified as normal weight (<25 kg/m^2^), overweight (25 to <30 kg/m^2^), and obesity (≥30 kg/m^2^); history of diabetes mellitus (fasting glucose level ≥126 mg/dL), hypertension (resting blood pressure ≥140/90 mmHg), CVD, cancer, chronic lung disease, and renal disease (urine albumin-to-creatinine ratio ≥ 30 mg/g) were ascertained using the self-reported doctor’s diagnosis, medication use, and/or laboratory-measured biochemical indicators. (3) Lifestyle factors: physical activity (inactive (has no leisure-time physical activity), insufficient (1~5 times/week for moderate activity with METs ranging from 3~6 or 1~3 times/week for vigorous activity with METs > 6), as well as active (has more leisure-time moderate or vigorous activity than above mentioned)); smoking status (individuals who reported smoking <100 cigarettes over lifetime were coded as never smokers, those who had smoked ≥100 cigarettes but quit smoking as past smokers, and those who still smoked now as current smokers); drinking status (never drinker, abstainer (defined as >12 drinks in any given year in life but without alcohol consumption in the past 12 months), and current drinker (defined as having as least 1 drinks in past 12 months)). In addition, whole-blood biochemical markers, measured in NHANES Laboratory, such as lead and C-reactive protein (CRP), were also considered in the sensitivity analysis.

### 2.5. Statistical Analysis

Appropriate sampling weights were taken into considerations in all analyses for the complex, multistage sample design implemented by NHANES. As recommended by the Endocrine Society Clinical Practice Guidelines, all included OA patients were classified into four categories, that is, serum 25(OH)D < 25.0 nmol/L was regarded as severe deficiency, 25.0–49.9 nmol/L as moderate deficiency, 50.0–74.9 nmol/L as insufficient, and ≥75.0 nmol/L as sufficient [16]. A descriptive analysis was followed to evaluate between-group differences in baseline characteristics, and χ^2^ tests were used for categorical variables, whereas generalized linear regressions were used for continuous variables.

Before fitting the Cox regression model, tests based on Schoenfeld residuals were initially incorporated for the proportional hazards (PHs) assumption. The results suggested that two predictors (i.e., physical activity and PIR) did not satisfy the PH assumption in several models. In those cases, we employed the stratified Cox model instead. Estimated effects size was presented as hazard ratios (HRs) and corresponding 95% confidence intervals (CIs). Serum 25(OH)D concentration, as the independent variable, was modeled categorically (four categories) and continuously (log-transformed value), respectively. In Model 1, age, gender, and race/ethnicity were adjusted; Model 2 further adjusted (from Model 1) for education level, BMI, PIR (when applicable), physical activity (when applicable), smoking status, and drinking status. In addition, Model 3 further adjusted (from Model 2) for diabetes mellitus, hypertension, CVD, cancer, chronic lung disease, and renal disease. Furthermore, the non-linear association of interest (on a continuous scale) was examined by the restricted cubic spline regression models with full adjustment for confounders.

Stratified analyses, only for all-cause mortality with a sufficient number of deaths, were also performed to explore underlying effect modification, including age (≤60 years, >60 years), gender, race/ethnicity (non-Hispanic white, others), BMI (<30 kg/m^2^, ≥30 kg/m^2^), smoking status (never smoker, current smoker, past smoker), drinking status (never drinker, current drinker, abstainer), physical activity (inactive, insufficient, or active), chronic lung disease (yes, no), and renal disease (yes, no). The potential interactions between serum 25(OH)D concentrations and these stratifying variables were detected by adding their cross-product terms into the model accordingly.

In addition, to check the robustness of the association, a series of sensitivity analyses were performed as follows: (1) considering that cancer often leads to worse survival, we excluded subjects with a history of cancer to avoid a situation in which cancer patients were unevenly distributed across comparison groups; (2) likewise, analyses were conducted after the exclusion of OA patients with a history of CVD; (3) we also removed those whose deaths occurred during the first two years of follow-up to avoid potential reverse causality as much as possible; (4) history of fracture, lipid profiles (i.e., the ratio of total cholesterol to HDL) [17], CRP [18], and blood lead [19], thought to likely influence the observed relationships according to suggestive biological links, were separately adjusted in the model.

All analyses were carried out in Stata/SE version 15.1 and R version 4.0.5.

## 3. Results

### 3.1. Baseline Characteristics

There were 4570 OA patients included in this analysis, of which approximately 65.53% were female. The mean age of all included patients was 61.82 ± 12.94 years old. The geometric mean (95% CI) of serum 25(OH)D concentration was 63.90 (95% CI: 63.08–64.73) nmol/L; a total of 16.54% of the patients were Vitamin D deficient (<50 nmol/L), and 48.15% were Vitamin D insufficient (<75 nmol/L). By the census day of 31 December 2019, 1388 OA patients died (median follow-up: 7.6 (interquartile range: 4.2–12.3) years), including 427 (30.8%) from CVD, 268 (19.3%) from cancer, and 693 (49.9%) from other causes.

Table 1 presents the baseline information of OA patients by serum 25(OH)D status. OA patients with higher levels of serum 25(OH)D tended to be older, non-Hispanic whites, as well as well educated. They were less likely to be overweight/obese, be current smokers, or have comorbidities such as diabetes, cancer, and renal disease. Regarding biochemical markers, patients with higher serum 25(OH)D concentrations tended to have higher HDL and CRP. We did not observe significant differences in LDLs and triglycerides.

### 3.2. Vitamin D and Mortality

The associations of serum 25(OH)D concentrations (on a continuous scale) with mortality risk can be visualized in Figure 2. An L-shaped association was found for all-cause mortality, and the concentration of serum 25(OH)D related to the lowest all-cause mortality risk was approximately 84.50 nmol/L. An inverse association was found for CVD-specific mortality.

The results based on serum 25(OH)D concentrations categories are summarized in Table 2. After multivariable adjustment, categories with higher levels of 25(OH)D were significantly related to low all-cause mortality and CVD mortality but not to cancer-specific mortality. The multivariable-adjusted HRs (95% CIs) across four categories of serum 25(OH)D concentrations were 1.00 (reference), 0.49 (0.31, 0.75), 0.45 (0.29, 0.68), and 0.43 (0.27, 0.69) for all-cause mortality and 1.00 (reference), 0.28 (0.14, 0.59), 0.25 (0.12, 0.51), and 0.24 (0.11, 0.49) for CVD-specific mortality.

### 3.3. Stratified and Sensitivity Analyses

The results of the stratified analysis are shown in Table 3. Such inverse associations remained when analyses were stratified by age (≤60 or >60 years), gender (male or female), race/ethnicity (non-Hispanic white or other), BMI (<30 or ≥30 kg/m^2^), smoking status (never, current, or past), drinking status (never, current, or abstainer), physical activity (inactive or insufficient/active), chronic lung disease (yes or no), renal disease (yes or no), and any type of comorbidities (yes or no), even though in some strata, the associations became insignificant. Of note, no significant interactions (*p* _interaction_ > 0.05) were found between serum 25(OH)D and stratum variables except for the variable of any type of comorbidities (yes or no).

To further test the robustness of our findings, several sensitivity analyses were performed. Sensitivity analyses excluding OA patients with a history of cancers (*n* = 930), patients with a history of CVD (*n* = 934), and patients who died within two years of follow-up (*n* = 182) showed similar results (Appendix A). When further adjusted for fractures (yes or no, *n* = 3480), lipid profiles (*n* = 4525), CRP (*n* = 3440), or lead (*n* = 4023), the results did not change largely (Appendix A).

## 4. Discussion

In this study of 4570 OA patients from the NHANES III and NHANES 2001–2018 cohorts, after a median follow-up of 7.6 (interquartile range: 4.2–12.3) years, an L-shaped association was found for all-cause mortality, and the concentration of serum 25(OH)D related with the lowest all-cause mortality risk was approximately 84.50 nmol/L. An inverse association was found for CVD-specific mortality. Additionally, the results based on categories of 25(OH)D concentrations indicated that OA patients with sufficient serum 25(OH)D were associated with a decreased risk of all-cause and CVD mortality compared with those with lower 25(OH)D levels; however, such associations were not significant for cancer-specific mortality. Similar results were found when the analyses were stratified by factors including age, race/ethnicity, BMI, smoking status, drinking status, physical activity, and comorbidities. Intriguingly, when the analysis was stratified by gender, relatively lower HRs were observed in men than in women. No profound changes were observed in the sensitivity analyses, suggesting that our results were robust.

Our findings elucidated a protective role of Vitamin D in reducing mortality among OA patients, which is in line with previous studies concentrating on the general population and diabetic patients [11,12]. Although Vitamin D is considered as a highly influential factor in OA, only a few studies found a protective effect on OA onset and progression [8,20,21]. Of note, studies that did not support a preventive role of Vitamin D were conducted mainly among the population with higher 25(OH)D status (≥50 nmol/L) [6,7,22]; it is, therefore, possible that the use of an inappropriate population might conceal the true effect of Vitamin D [23]. Our study, however, found a beneficial effect of using Vitamin D on the long-term outcome, i.e., mortality reduction. In addition, by making rough comparisons, we found that the HRs reported in our analysis were slightly smaller than those among diabetic patients when similar covariates were adjusted. This finding indicated that OA patients might benefit more from higher Vitamin D levels than diabetic patients, thus highlighting the significance of sufficient Vitamin D intake in OA patients.

Bone diseases are a major cause of death in the elderly [24], and it is widely known that vitamin D helps the body to absorb and use calcium, protecting the bones [25]. Serum vitamin D is first converted into hormone calcitriol (known as “active Vitamin D”), which can promote the absorption of calcium, for example, by acting on the nuclei of small intestinal mucosal cells [26]. Calcium is subsequently combined with other minerals to form hard crystals, giving bones strength and structure. Additionally, calcium also has benefits in other body systems, such as reducing blood pressure and cholesterol levels, preventing tooth loss, and etc. [27]. The benefits mentioned above, whether relative to the bones, or endocrine or dental, have an essential impact on the reduction in mortality in later life. On the contrary, in the situation of Vitamin D deficiency, the body has to take calcium from its stores in the skeleton. Weakened existing bone leads to a less favorable survival [24].

Despite the benefit of Vitamin D in mortality reduction, a ceiling effect was identified via the restricted cubic spline analysis. An L-shaped association was found between serum 25(OH)D concentrations and all-cause mortality, and such a finding was consistent with that of Fan et. al., where the study population was the general population (regardless of disease history) from a large cohort, UK Biobank [11]. Nevertheless, they reported that the concentration of 25(OH)D associated with the lowest risk of all-cause mortality was 60 nmol/L [11], which was lower than that in our study (84 nmol/L). A possible reason could be the different populations used (general population vs. OA patients). Regarding the association with CVD mortality, an inverse trend was found in our analysis. Related studies focused on OA patients are scarce; however, several studies focused on the general population or prediabetic patients showed a similar decreasing trend without stating a ceiling effect, which is in line with our finding [28,29]. However, the conclusion on this issue is inclusive. For instance, a reverse J-shaped association was observed in CopD Study, and the 25(OH)D level of 70 nmol/L was related with the lowest CVD mortality risk [30]. Moreover, Xiao et al. recently published a study that also focused on the NHANES database [31]; they observed an L-shaped association between serum 25(OH)D concentrations and CVD and all-cause mortality, and the concentration of 54.40 nmol/L was related with the lowest all-cause mortality risk. The disparities between their findings and our ours could be partly explained by the follow-up period and population sample size, as our analysis was based on newly released data, NHANES 2001–2018, which included more OA patients (4570 vs. 2556) and had a longer follow-up time.

Although the associations in several subgroups did not reach statistical significance due to limited sample size, in general, the results of stratified analyses were in line with that of our main analysis. Of note, when the analysis was stratified by gender, relatively lower HRs were observed in men than in women, which suggested men might benefit more from higher Vitamin D status than women in terms of mortality reduction. Such findings can be possibly explained by gender differences in the pathogenesis and treatment of OA. Studies showed that women tended to experience a higher prevalence and more severe OA (such as decreased cartilage volume, experiencing more clinical pain, and physical difficulty) than men [32]. More importantly, most female patients included in our study were postmenopausal women that suffered hormonal changes, especially decreasing estrogen levels [33]. In such a population, higher Vitamin D exclusively might not be able to achieve benefits comparable to those in men, and more gender-specific factors such as an adequate estrogen level might also of importance [34,35]. The identification of gender-based differences in mortality reduction would be beneficial for the development of gender-personalized therapeutic strategies for OA patients, especially for postmenopausal women.

One of the major strengths in the current study is that a nationally representative sample of OA patients in the U.S. with long follow-up was used. Second, information on the cause of death was acquired via the linkage to National Death Index records. Moreover, the comprehensive data from NHANES allowed us to adjust a wide range of confounders such as lifestyle factors, socioeconomic status, race/ethnicity, and comorbidities.

Several limitations of our study should be noted. Firstly, due to the lack of repeated Vitamin D measurements, we could not ascertain the associations of dynamic Vitamin D status with mortality. Secondly, the inclusion of OA patients was based on a simple question, “Doctors ever told you had OA”, without further confirmation, using participants’ medical records. However, we did not anticipate that such bias could influence our results significantly. Another limitation in our study was the limited statistical power for detecting the associations between 25(OH)D and cancer-specific mortality, given the low frequency of cancer deaths. Additionally, studies showed that the role of Vitamin D in cancer incidence and survival and in different sites of cancer might be different [36,37,38]. Therefore, when feasible, future studies that evaluate the effect of Vitamin D on different sites of cancer are warranted. Finally, due to the study design, we could not directly deduce the causality of Vitamin D with mortality in OA patients. Theoretically, this question can be addressed with the utilization of randomized controlled trials and Mendelian randomization analyses, which unfortunately we were not able to perform in this study. Future studies that evaluate the potential casual associations are of vital use in elucidating the effect of Vitamin D on a long-term basis (such as mortality reduction) on OA patients.

## 5. Conclusions

This study elucidated that OA patients with sufficient serum 25(OH)D were significantly associated with a decreased risk of all-cause and CVD mortality than those at lower 25(OH)D levels, which suggested a beneficial role of Vitamin D on a long-term basis. In addition, relatively lower HRs were observed in men than in women, highlighting the need of developing gender-personalized therapeutic strategies for OA patients, especially postmenopausal women.

## Figures and Tables

**Figure 1 nutrients-14-04629-f001:**
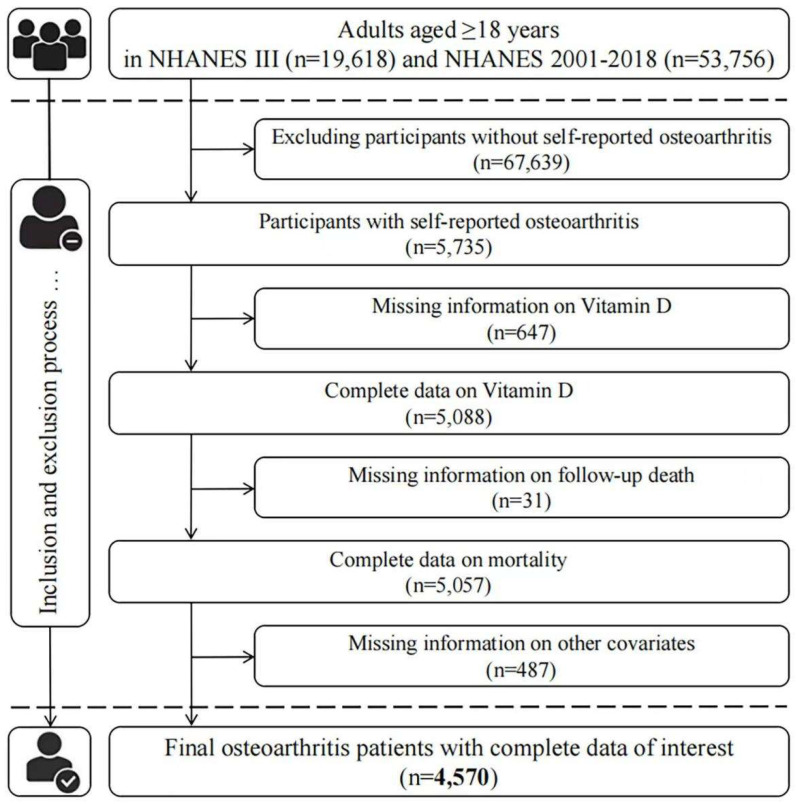
Flow chart of patient inclusion.

**Figure 2 nutrients-14-04629-f002:**
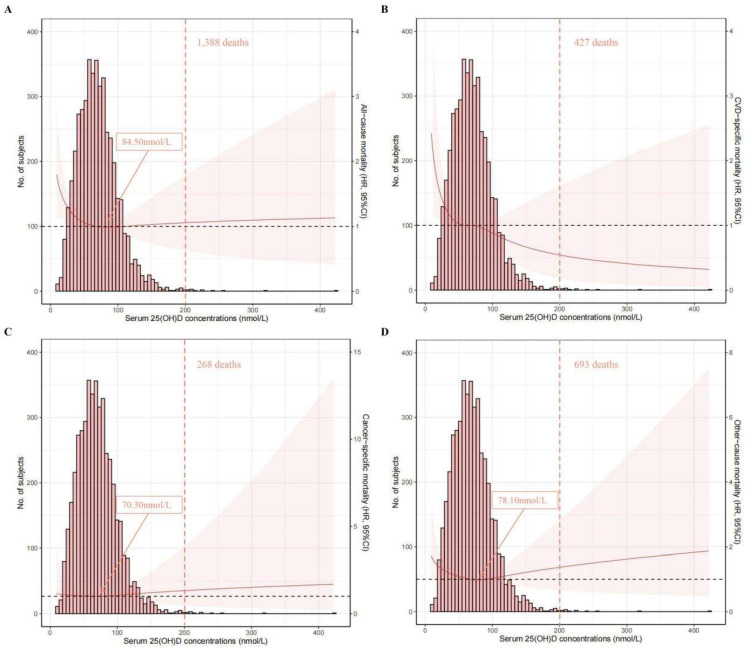
Spline analyses of all-cause (**A**), CVD-specific (**B**), cancer-specific (**C**), and other-cause (**D**) mortality according to serum 25(OH)D concentrations, with the frequency distribution histogram in the background. Solid red lines represent multivariable-adjusted hazard ratios, with light-red areas showing the 95% confidence intervals derived from restricted cubic spline regressions (spline analyses were adjusted for age, gender, race/ethnicity, education, body mass index, physical activity, smoking status, drinking status, diabetes mellitus, hypertension, cardiovascular disease, cancer, chronic lung disease, and renal disease); arrows indicate the concentration of 25(OH)D with the lowest risk of mortality; over 99% of the participants were located to the left of the red dotted lines.

**Table 1 nutrients-14-04629-t001:** Basic characteristics of participants with OA according to serum 25(OH)D concentrations in NHANES III and NHANES 2001–2018.

Characteristic	Serum 25(OH)D Concentrations (nmol/L) ^a^				
	<25.0 (*n* = 136)	25.0–49.9 (*n* = 1054)	50.0–74.9 (*n* = 1579)	≥75.0 (*n* = 1801)	*p*
Age (years, mean ± SD)	60.89 ± 13.54	59.50 ± 13.96	59.89 ± 13.49	63.70 ± 11.95	<0.001
Gender, *n* (%)					
Male	31 (15.40)	355 (32.96)	642 (40.57)	591 (31.78)	<0.001
Female	105 (84.60)	699 (67.04)	937 (59.43)	1210 (68.22)	
Race/ethnicity, *n* (%)					
Non-Hispanic white	45 (53.21)	558 (71.70)	1068 (81.44)	1392 (88.78)	<0.001
Non-Hispanic black	57 (26.47)	255 (13.15)	190 (5.48)	145 (3.29)	
Mexican American	22 (11.86)	137 (5.68)	121 (2.69)	89 (1.51)	
Others	12 (8.45)	104 (9.47)	200 (10.38)	175 (6.41)	
Education level, *n* (%)					
Below high school	49 (27.09)	325 (19.84)	359 (13.54)	310 (10.32)	<0.001
High school or equivalent	35 (22.50)	270 (28.13)	391 (24.78)	420 (22.70)	
Above high school	52 (50.40)	459 (52.03)	829 (61.68)	1071 (66.98)	
Family poverty income ratio, *n* (%)					
≤1	33 (18.19)	192 (15.49)	218 (9.40)	194 (7.59)	<0.001
1< to ≤3	58 (45.14)	465 (39.33)	645 (34.74)	725 (31.37)	
>3	45 (36.68)	397 (45.19)	716 (55.85)	882 (61.04)	
Body mass index, *n* (%)					
Normal	18 (10.72)	177 (15.42)	340 (20.93)	475 (22.74)	<0.001
Overweight	36 (27.67)	326 (30.33)	495 (29.01)	609 (33.86)	
Obese	82 (61.62)	551 (54.26)	744 (50.06)	717 (43.40)	
Physical activity, *n* (%)					
Inactive	112 (84.55)	675 (59.63)	816 (46.96)	962 (48.79)	<0.001
Insufficient	7 (3.90)	157 (15.05)	296 (19.64)	259 (15.84)	
Active	17 (11.55)	222 (25.32)	467 (33.39)	580 (35.37)	
Smoking status, *n* (%)					
Never smoker	66 (37.58)	506 (45.41)	730 (44.55)	890 (51.50)	<0.001
Current smoker	35 (30.94)	216 (24.13)	242 (15.35)	220 (13.40)	
Past smoker	35 (31.49)	332 (30.46)	607 (40.10)	691 (35.09)	
Drinking status, *n* (%)					
Never drinker	30 (18.71)	188 (13.48)	226 (10.23)	221 (8.37)	<0.001
Current drinker	74 (63.13)	574 (59.82)	1065 (72.22)	1229 (73.97)	
Abstainer	32 (18.16)	292 (26.70)	288 (17.55)	351 (17.67)	
Comorbidities, *n* (%)					
Diabetes mellitus	44 (24.23)	312 (26.38)	363 (20.01)	373 (19.44)	<0.001
Hypertension	88 (67.56)	693 (61.84)	1006 (58.90)	1229 (63.56)	0.049
Cardiovascular disease	29 (24.87)	226 (18.75)	293 (18.16)	386 (18.87)	0.154
Cancer	15 (12.44)	178 (19.57)	299 (19.69)	438 (25.67)	<0.001
Chronic lung disease	31 (33.28)	147 (12.93)	207 (13.79)	219 (10.11)	0.004
Renal disease	38 (28.21)	224 (18.18)	271 (13.05)	277 (13.10)	<0.001
Whole-blood biochemical markers (GM, 95% CI)					
Total cholesterol (*n* = 4529), mmol/L	5.130 (1.941, 5.325)	5.168 (5.096, 5.240)	5.083 (5.030, 5.137)	5.003 (4.952, 5.053)	<0.001
HDL (*n* = 4526), mmol/L	1.349 (1.283, 1.417)	1.297 (1.274, 1.320)	1.325 (1.305, 1.344)	1.419 (1.399, 1.439)	<0.001
LDL (*n* = 2149), mmol/L	2.651 (2.431, 2.891)	2.980 (2.893, 3.071)	2.925 (2.856, 2.997)	2.772 (2.711, 2.835)	0.002
Triglyceride (*n* = 4549), mmol/L	1.306 (1.144, 1.491)	1.436 (1.377, 1.497)	1.391 (1.341, 1.443)	1.320 (1.275, 1.367)	0.016
CRP (*n* = 3440), mg/dL	0.423 (0.341, 0.526)	0.334 (0.311, 0.359)	0.269 (0.253, 0.287)	0.230 (0.215, 0.245)	<0.001
Lead (*n* = 4023), umol/L	0.105 (0.090, 0.122)	0.095 (0.090, 0.099)	0.080 (0.077, 0.083)	0.068 (0.066, 0.071)	<0.001

NHANES, National Health and Nutrition Examination Survey; 25(OH)D, 25-Hydroxyvitamin D; OA, osteoarthritis; HDL, high-density lipoprotein; LDL, low-density lipoprotein; CRP, C-reactive protein; GM, geometric mean; CI, confidence interval; *n*, number of subjects; SD, standard deviation; %, weighted percentage. ^a^ Vitamin D status was categorized into four groups, severe deficiency (<25.0 nmol/L), moderate deficiency (25.0–49.9 nmol/L), insufficient (50.0–74.9 nmol/L), and sufficient (≥75.0 nmol/L), according to the Endocrine Society Clinical Practice Guidelines. All estimates accounted for complex survey designs.

**Table 2 nutrients-14-04629-t002:** HRs (95% CIs) for all-cause and cause-specific mortality according to serum 25(OH)D concentrations among OA patients in NHANES III and NHANES 2001–2018.

	Serum 25(OH)D Concentrations (nmol/L)				Per One-Unit Increment in Natural Log-Transformed 25(OH)D
	<25.0	25.0–49.9	50.0–74.9	≥75.0	
All-cause mortality					
Number of deaths/total	58/136	429/1054	510/1579	391/1801	
Model 1 ^a^	Ref.	0.43 (0.27, 0.67)	0.36 (0.23, 0.56)	0.32 (0.20, 0.53)	0.66 (0.53, 0.83)
Model 2 ^b^	Ref.	0.46 (0.30, 0.71)	0.40 (0.26, 0.62)	0.39 (0.24, 0.63)	0.76 (0.61, 0.94)
Model 3 ^c^	Ref.	0.49 (0.31, 0.75)	0.45 (0.29, 0.68)	0.43 (0.27, 0.69)	0.81 (0.65, 1.00)
CVD mortality					
Number of deaths	22	138	156	111	
Model 1 ^a^	Ref.	0.27 (0.12, 0.57)	0.20 (0.09, 0.41)	0.17 (0.08, 0.38)	0.47 (0.30, 0.73)
Model 2 ^b^	Ref.	0.27 (0.13, 0.57)	0.23 (0.11, 0.47)	0.22 (0.10, 0.48)	0.59 (0.37, 0.95)
Model 3 ^c^	Ref.	0.28 (0.14, 0.59)	0.25 (0.12, 0.51)	0.24 (0.11, 0.49)	0.61 (0.40, 0.94)
Cancer mortality					
Number of deaths	8	75	96	89	
Model 1 ^a^	Ref.	0.70 (0.29, 1.73)	0.56 (0.23, 1.37)	0.65 (0.27, 1.59)	0.85 (0.58, 1.26)
Model 2 ^b^	Ref.	0.77 (0.31, 1.91)	0.69 (0.27, 1.73)	0.83 (0.34, 2.03)	1.00 (0.70, 1.43)
Model 3 ^c^	Ref.	0.74 (0.29, 1.87)	0.68 (0.26, 1.73)	0.81 (0.32, 2.04)	1.00 (0.71, 1.42)
Other mortality					
Number of deaths	28	216	258	191	
Model 1 ^a^	Ref.	0.52 (0.30, 0.90)	0.47 (0.28, 0.78)	0.39 (0.23, 0.68)	0.73 (0.57, 0.94)
Model 2 ^b^	Ref.	0.56 (0.32, 0.98)	0.51 (0.31, 0.86)	0.45 (0.26, 0.78)	0.80 (0.63, 1.02)
Model 3 ^c^	Ref.	0.64 (0.35, 1.15)	0.61 (0.36, 1.03)	0.54 (0.30, 0.96)	0.88 (0.68, 1.14)

NHANES, National Health and Nutrition Examination Survey; 25(OH)D, 25-Hydroxyvitamin D; OA, osteoarthritis; CVD, cardiovascular disease; HR, hazard ratio; CI, confidence interval; Ref., reference. ^a^ Model 1 adjusted for age, gender, and race/ethnicity; ^b^ Model 2 further adjusted (from Model 1) for education, body mass index, physical activity, smoking status, and drinking status; ^c^ Model 3 further adjusted (from Model 2) for diabetes mellitus, hypertension, cardiovascular disease, cancer, chronic lung disease, and renal disease.

**Table 3 nutrients-14-04629-t003:** Stratified analyses of the associations (HRs, 95% CIs) between serum 25(OH)D concentrations and all-cause mortality among OA patients in NHANES III and NHANES 2001–2018.

	Serum 25(OH)D Concentrations (nmol/L)				*p* _interaction_ ^b^
	<25.0 ^a^	25.0–49.9 ^a^	50.0–74.9 ^a^	≥75.0 ^a^	
Age (years)					0.46
≤60 (*n* = 1624)	Ref.	0.13 (0.07, 0.25)	0.15 (0.08, 0.26)	0.19 (0.08, 0.43)	
>60 (*n* = 2946)	Ref.	0.71 (0.40, 1.24)	0.58 (0.33, 1.03)	0.57 (0.31, 1.04)	
Gender					0.83
Male (*n* = 1619)	Ref.	0.22 (0.09, 0.54)	0.20 (0.08, 0.50)	0.21 (0.08, 0.52)	
Female (*n* = 2951)	Ref.	0.58 (0.33, 1.02)	0.53 (0.31, 0.91)	0.51 (0.28, 0.92)	
Race/ethnicity					0.32
Non-Hispanic white (*n* = 3063)	Ref.	0.46 (0.26, 0.82)	0.44 (0.26, 0.75)	0.44 (0.24, 0.79)	
Others (*n* = 1507)	Ref.	0.56 (0.31, 1.01)	0.44 (0.23, 0.83)	0.28 (0.14, 0.56)	
Body mass index					0.63
<30 (*n* = 2476)	Ref.	0.67 (0.36, 1.26)	0.71 (0.38, 1.32)	0.61 (0.32, 1.17)	
≥30 (*n* = 2094)	Ref.	0.35 (0.20, 0.61)	0.28 (0.15, 0.50)	0.33 (0.17, 0.62)	
Smoking status					0.12
Never smoker (*n* = 2192)	Ref.	0.66 (0.38, 1.14)	0.68 (0.40, 1.17)	0.68 (0.39, 1.21)	
Current smoker (*n* = 713)	Ref.	0.18 (0.07, 0.45)	0.19 (0.07, 0.54)	0.16 (0.06, 0.43)	
Past smoker (*n* = 1665)	Ref.	0.57 (0.23, 1.46)	0.43 (0.17, 1.08)	0.42 (0.17, 1.03)	
Drinking status					0.27
Never drinker (*n* = 665)	Ref.	0.70 (0.29, 1.68)	0.70 (0.30, 1.64)	0.41 (0.17, 1.00)	
Current drinker (*n* = 2942)	Ref.	0.30 (0.14, 0.62)	0.27 (0.13, 0.57)	0.29 (0.13, 0.66)	
Abstainer (*n* = 963)	Ref.	0.83 (0.41, 1.70)	0.68 (0.31, 1.47)	0.64 (0.29, 1.40)	
Physical activity					0.98
Inactive (*n* = 2565)	Ref.	0.53 (0.30, 0.93)	0.50 (0.29, 0.87)	0.52 (0.28, 0.98)	
Insufficient or active (*n* = 2005)	Ref.	0.38 (0.18, 0.79)	0.32 (0.15, 0.68)	0.29 (0.13, 0.63)	
Chronic lung disease					0.73
Yes (*n* = 604)	Ref.	0.45 (0.15, 1.34)	0.33 (0.10, 1.09)	0.28 (0.08, 0.99)	
No (*n* = 3966)	Ref.	0.48 (0.31, 0.75)	0.46 (0.29, 0.72)	0.44 (0.27, 0.71)	
Renal disease					0.18
Yes (*n* = 810)	Ref.	0.71 (0.32, 1.58)	0.78 (0.35, 1.70)	0.68 (0.31, 1.49)	
No (*n* = 3760)	Ref.	0.42 (0.24, 0.73)	0.37 (0.21, 0.65)	0.37 (0.19, 0.71)	
Any type of comorbidities					<0.001
Yes (*n* = 3704)	Ref.	0.53 (0.32, 0.88)	0.47 (0.29, 0.77)	0.44 (0.25, 0.77)	
No (*n* = 866)	Ref.	0.08 (0.03, 0.24)	0.11 (0.04, 0.30)	0.11 (0.03, 0.34)	

NHANES, National Health and Nutrition Examination Survey; 25(OH)D, 25-Hydroxyvitamin D; OA, osteoarthritis; HR, hazard ratio; CI, confidence interval; *n*, number of subjects; Ref., reference. ^a^ Model adjusted for age, gender, race/ethnicity, education, body mass index, physical activity, smoking status, drinking status, diabetes mellitus, hypertension, cardiovascular disease, cancer, chronic lung disease, and renal disease. ^b^ Wald test was performed to examine the interaction between continuous serum 25(OH)D and stratification variables.

## Data Availability

The data used in this study are freely available for download by the public at: https://wwwn.cdc.gov/nchs/nhanes/Default.aspx (accessed on 31 August 2022).

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
