# Peer review of "Vitamin D Status and Risk of All-Cause and Cause-Specific Mortality in Osteoarthritis Patients: Results from NHANES III and NHANES 2001–2018"

_nutrients, 2022, doi:10.3390/nu14214629_

Round 1

Reviewer 1 Report

This is a study on the effect of vitamin D in the serum on the survival rate among patients with osteoarthritis. The study included 4,570 self-reported OA patients, including 1,388 death in the study period. While the same study has been conducted for the general population, this is the first study focusing on OA patients. The manuscript is written very well with a clear introduction, study design, results, and discussion including the comparison with other studies. Here are several minor comments to improve the current manuscript.

Minor

·       CVD: in the abstract, the abbreviation of CVD needs to be explained.

·       Cancer-specific mortality: the study did not identify a linkage between VD levels and cancer-specific mortality. Is this expected? In the study, 30% of participants (1,388) died in the study period and it seems that the number of cancer death is close to the number of CVD death. What is the expected sample size to detect any effect based on the data for the general population?

·       Methodological discrepancy: It is recommended that this discrepancy is described quantitatively and the possible effects on the results are estimated.

·       Calcium balance and bone health: the study is introduced in connection with the importance of the role of VD in calcium balance and bone health. However, the study is not well connected to calcium balance or bone health. It is recommended to add a description in the discussion regarding the mortality linked to calcium balance and bone health.

Reviewer 2 Report

The objective of   this study was we to evaluate  whether serum 25(OH)D concentrations were associated with all-cause and cause-specific  mortality in OA patients from the NHANES III and the newly released NHANES 2001- 2018 database.  This study elucidated that OA patients with sufficient serum 25(OH)D were significantly associated with decreased risks of all-cause and CVD mortality than those at lower  25(OH)D levels, which suggested a beneficial role of Vitamin D on a long-term basis.

The introduction is correct considering the prevalence of osteoarthritis, the possible role of vitamin D in its pathophysiology and the lack of mortality data in this population. The bibliographic references are  appropriates

The methodology is clearly described, with an adequate sample size and a broad description of the statistical analysis. The results, although extensive, are clearly expressed and easy to understand The discussion is correct adjusting to the results obtained
